# Electronic structure of CeAuAl$_3$ using density functional theory

**André Deyerling$^{1\star}$, Marc A. Wilde$^1$ and Christian Pfleiderer$^{1,2,3}$**

**1** Physik Department, Technische Universität München, D-85748 Garching, Germany
**2** MCQST, Technische Universität München, D-85748 Garching, Germany
**3** Centre for Quantum Engineering (ZQE), Technische Universität München, D-85748 Garching, Germany

$\star$ andre.deyerling@tum.de

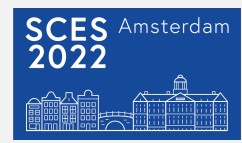

## Abstract

We studied the magnetic properties and electronic structure of CeAuAl$_3$ using density functional theory. This compound shows a large Sommerfeld coefficient, a Kondo temperature, T$_K$ = 4 K [1] and antiferromagnetic order below T$_N$ = 1.1 K [2]. We calculated the magnetic groundstate of CeAuAl$_3$ and the magnetic anisotropy energies. Treating the 4f-electrons as localized with DFT+U we obtain a good match with the magnetic properties observed experimentally. We also report salient features of the electronic structure of CeAuAl$_3$, including features of the Fermi surface and associated quantum oscillatory spectra, when the 4f-electrons are treated either as localized or itinerant.

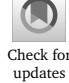

## 1 Introduction

Cerium based compounds have attracted significant attention due to strong competing energy scales that lead to heavy fermion behavior, magnetic order, superconductivity and magneto-elastic coupling. For the family of CeTAl$_3$ compounds (T=Ag, Au, Cu, Pd, Pt) different forms of magnetic order [3] and coupling between crystal field states and phonons have generated great interest in recent years [4,5]. The physical properties are tightly linked to the 4f-electrons and their interaction with the conduction electrons. In order to obtain a better understanding of the hybridization between the f- and conduction electrons a detailed study of the electronic structure of these compounds is required. An important concept when discussing the electronic structure is the observation of a small or large Fermi surface according to the Luttinger theorem [6,7] and it's relation to the localized or itinerant nature of 4f-electrons. The small or large Fermi surface are often found in different parts of the field-temperature-pressure phase diagram [8–10]. Relatively unexplored so far is the electronic structure of the series of CeTAl$_3$ compounds, both theoretically or experimentally. We present calculations of the magnetic properties and electronic structure of CeAuAl$_3$ as obtained using density functional theory.

## 2 Computational methods

We used the full potential APW+lo method as implemented in WIEN2k [11] and Elk [12]. WIEN2k was used to calculate the properties for collinear magnetic order in CeAuAl$_3$ whereas Elk was used for helical non-collinear magnetic structures. For the latter the generalized Bloch theorem was employed which allows to calculate magnetic structures with long wavelength modulations within the chemical unit cell [13]. The itinerant 4f-electron picture was addressed by DFT calculations whereas the properties assuming a localized 4f-electron were calculated by DFT+U. For the Hubbard correction in DFT+U the implementation of Liechtenstein et al. [14] was used in the fully localized limit and $J$ was set to zero. In DFT+U calculations multiple local minima corresponding to different occupations of the 4f-states may occur [15]. To determine the global minimum multiple local minima were explored by starting with a fixed 4f-density matrix and relaxing it in a second calculation. The lowest energy solution was taken as the global minimum. For the electronic structure and magnetic properties presented in the following, the room temperature experimental lattice constants were used [16]. No significant changes were observed when using optimized lattice constants.

For the DFT calculations the exchange correlation functionals LDA by Perdew and Wang [17], PBE [18] and PBEsol [19] were used. In WIEN2k the muffin tin radii $R_{MT}$ were set to 2.21 b for Ce, 2.30 b for Au and 2.00 b for Al and $RK_M$ was set to 9.5. For the helical magnetic order calculated in Elk the muffin tin radii $R_{MT}$ were set to 2.50 b for Ce, 2.50 b for Au and 2.10 b for Al and $RK_M$ was set to 9.0. The results are well converged for $RK_M \geq 9.0$. The cut-off energy between valence and core electrons is $-8.3$ Ry. Different k-grids were used for specific physical properties. In WIEN2k and Elk different symmetry settings are used for the unit cells and their k-grids. Therefore, we specifically report the unit cell and the DFT package for each k-grid. Structure optimization was done with a 21x21x21 grid (primitive cell, WIEN2k), the electronic and magnetic properties with a 36x36x36 grid (primitive cell, WIEN2k) and the helical magnetic structure with a 27x27x20 grid (primitive cell, Elk).

## 3 Results

Conflicting terminology is sometimes used when discussing the character of 4f-electrons in heavy fermion systems. In the following we distinguish the following notions: localized magnetism, localized 4f-electron, and itinerant 4f-electron. Localized magnetism is used in the following in the sense that the magnetic properties of a 4f-electron compound are well described without major contributions from a hybridization of the f-electrons with the conduction electrons such as the Kondo effect. With localized 4f-electrons and itinerant 4f-electrons we refer to the large and small Fermi surfaces, respectively, in the sense of Luttingers theorem. Localized magnetism is always found for localized 4f-electrons but not vice versa. In the following sections we will show that the experimental findings and our own ab initio calculations are consistent with the picture of localized magnetism. We will then give predictions for the large and the small Fermi surface in this compound and comment on where in the magnetic phase diagram one or the other may be expected.

### 3.1 Physical properties

The known physical properties of CeAuAl$_3$ relevant to the character of the 4f-electron and to the applicability of DFT may be summarized as follows. CeAuAl$_3$ forms in a tetragonal crystal structure with space group I4mm, which lacks inversion symmetry [16]. Fermi liquid behavior [1] in combination with heavy fermion masses, have been reported for low temperatures,

with the heavy masses inferred from the Sommerfeld coefficient $\gamma$ of the electronic specific heat. Extracting $\gamma$ from measurements of the specific heat versus temperature is somewhat ambiguous in this compound due to the magnetic transition. One can either go to very low temperatures below the magnetic transition temperature as done by Paschen et al. [1] or focus on the specific heat above the magnetic transition temperature as done by Adroja et al. [2], resulting in $\gamma = 227, \frac{mJ}{molK^2}$ and $\gamma = 400 \frac{mJ}{molK^2}$ , respectively. The Kondo temperature is estimated to be around $4\,K$ based on the specific heat [1, 2], and around $30\,K$ based on the thermo-power [20]. Applying a magnetic field strongly suppresses the Sommerfeld coefficient $\gamma$ [2]. Around $B = 2\,T$ the Sommerfeld coefficient amounts to 1/4 of the zero field value and decreases to $\gamma \approx 50 \frac{mJ}{molK^2}$ at $B = 7\,T$.

The magnetic phase transition is a transition from a paramagnetic to an antiferromagnetic state which occurs at $T_N = 1.1\,K$ [1, 2, 20, 21]. The ordering vector $k$ of this structure was reported to be $(0, 0, 0.52)\,2\pi/c$ with an ordered moment at the Ce site of $1.05\,\mu_B$ [2]. At $9\,T$ a difference of $1.05\,\mu_B$ has been observed between the magnetisation along the magnetic easy axis which is parallel to the a-axis and the magnetic hard axis which is parallel to the c-axis [21]. Such a large anisotropy in the magnetisation suggests a prevalent influence of the crystal electric fields being associated with a localized 4f-electron as the origin of the magnetic properties. Also, the Curie-Weiss moment of $(2.55 - 2.66)\,\mu_B$ is consistent with the value expected for a free $Ce^{3+}$ ion [1, 21]. In inelastic neutron scattering magneto-elastic coupling between the first excited crystal field state and acoustic phonons has been reported [5]. In that study, the hybridization strength between the conduction- and the 4f-electrons was found to be very low for temperatures between 4K and 300K as compared to the localized 4f-electrons in $CeAl_2$ [22]. Thus, the experimental findings suggest that a large Fermi surface due to Kondo coupling may be expected in the regime where heavy fermion masses are observed and a small Fermi surface may be expected in the regime of lighter masses.

## 3.2 Ab initio results

The presentation of ab initio results is organized in three parts starting with the structural properties, followed by the magnetic properties and the electronic structure. First insights into the relevance of the localized electron picture may be inferred from the structural properties of $CeAuAl_3$. Structure optimization of the lattice constants and atomic positions was performed for different input parameters including the treatment of localized and itinerant 4f-electrons. Key results are summarized in table 1. The optimized crystal structures for

Table 1: Structural parameters of $CeAuAl_3$ as obtained from structure optimization within DFT. Unit cell volume $V$, lattice constants $a$ and $c$ and their ratio $c/a$ are compared to the high and low temperature structural parameters as observed in experiment. Good agreement with the experimental values for $T = 0.3\,K$ is obtained when 4f-electrons are treated as localized (LDA/PBE/PBEsol+U).

|  | $V$ ($Å^3$) | $a$ ($Å$) | $c$ ($Å$) | $c/a$ |
|---|---|---|---|---|
| Experiment (293.15 K) [16] | 204.026 | 4.337 | 10.850 | 2.502 |
| Experiment (0.30 K) [2] | 200.603 | 4.310 | 10.796 | 2.505 |
| PBE (non-magnetic) $U$=4 eV | 203.620 | 4.314 | 10.940 | 2.536 |
| PBEsol (spin polarized) $U$=4 eV | 201.753 | 4.310 | 10.861 | 2.520 |
| PBEsol (non-magnetic) $U$=4 eV | 196.436 | 4.260 | 10.824 | 2.536 |
| LDA (spin polarized) $U$=4 eV | 196.333 | 4.274 | 10.750 | 2.515 |
| LDA (spin polarized) $U$=0 | 193.022 | 4.262 | 10.629 | 2.494 |

localized 4f-electrons (table 1 lines 3-6) match experiment very well at low temperatures (table 1 line 2). The differences in volume are below 2.2% and the difference in $c/a$ below 1.3%. Comparing the localized description (table 1 lines 3-6) with the itinerant description (table 1 line 7), the localized description displays a better agreement between the calculated crystal structures and experiment as compared to the case of itinerant 4f-electrons. GGA+U (PBE, PBEsol) captures the unit cell volume $V$ and the size of the lattice constants $a$ and $c$ better than LDA+U. However, the ratio of the experimental lattice constants is best reproduced with LDA+U.

Comparison of the DFT results on the magnetic groundstate properties with experiment leads to the conclusion that localized magnetism accounts well for the properties of $CeAuAl_3$. The magnetic properties were calculated using the LDA functional as the GGA functionals in combination with DFT+U show a strong suppression of the orbital moment. This observation was earlier reported for fcc Cerium and appears to be a general characteristic of the GGA functionals [23]. We addressed the helical magnetic order of $CeAuAl_3$ as well as the spin polarized state corresponding to large applied magnetic fields. The latter also provides information on the magnetic anisotropy, because changing the direction of the collinear spin arrangement in the spin polarized state is similar to changing the direction of an applied magnetic field in experiment. The magnetic anisotropy energy shown in Table 2 and the size of the magnetic moments (Table 3) observed for the localized 4f-electron correspond to an easy axis along the $a$-axis and a hard axis along the $c$-axis, consistent with experiment. The size of the total magnetic moments parallel to the $a$- and $c$-axis is comparable to the spontaneous magnetic moment for magnetic fields along the $a$- and $c$-axis when extrapolating $B$ to zero [21]. The total magnetic moment is dominated by the orbital contribution. Using the generalized Bloch theorem we calculated the magnetic properties for the helical magnetic order. We find that only the localized 4f-electron picture (LDA+U) can stabilize a magnetic structure in agreement with experiment. Itinerant 4f-electrons lead to non-magnetic solutions. The energy differences between different magnetic ordering vectors for the localized 4f-electron are small which makes it necessary to use dense $k$-meshes and tight convergence criteria. Preliminary results are shown in figure 1. We conclude that the localized magnetism (DFT+U) captures the groundstate magnetic properties of $CeAuAl_3$ quite well.

Regarding the electronic structure of $CeAuAl_3$, to the best of our knowledge, no experimental studies of the bandstructure using, e.g., quantum oscillations or angle-resolved photoemis-

Table 2: Dependence of magnetic anisotropy energy on the degree of localization of the 4f-electron of $CeAuAl_3$ as parameterized by the Hubbard $U$.

| $U$ (eV) | 4f-electron | $E_{100} - E_{001}$ (meV) |
|---|---|---|
| 0 | (Itinerant) | -2.11 |
| 2 | (Localized) | -5.74 |
| 4 | (Localized) | -6.72 |

Table 3: Total magnetic moment at the Ce site of $CeAuAl_3$ for the $a$- and $c$-axis. The reference values from experiment are the spontaneous magnetic moment for magnetic fields along the $a$- and the $c$-axis when extrapolating $B$ to zero [21].

| Direction $\vec{m}$ | Itinerant | Localized | Experiment |
|---|---|---|---|
| [100] | $0.10\,\mu_B$ | $0.87\,\mu_B$ | $1.20\,\mu_B$ |
| [001] | $0.03\,\mu_B$ | $0.50\,\mu_B$ | $0.42\,\mu_B$ |

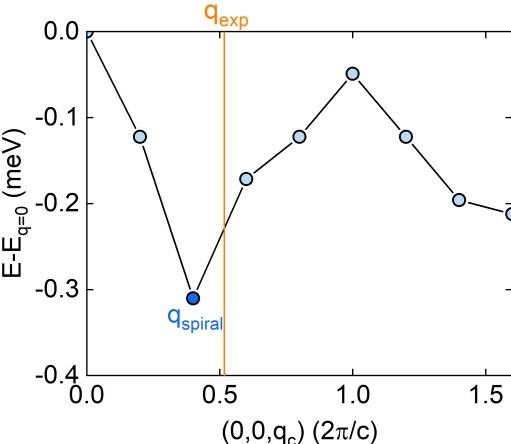

Figure 1: Difference in total energy of CeAuAl$_3$ when varying the magnetic ordering vector $q$ along the $c$-direction. The $q$-vector is given in units of the conventional reciprocal lattice vector. The calculated minimum at $q_{spiral}$ is close to the value $q_{exp}$ observed experimentally. The helical magnetic order is calculated using the generalized Bloch theorem [13] implemented in Elk [12] as the "spin spiral" method, which allows for calculations in the chemical unit cell, i.e., avoiding the use of supercells.

sion spectroscopy (ARPES) have been reported. As pointed out above, localized and itinerant 4f-electrons may exist in different parts of the magnetic phase diagram. In the following we show that the itinerant and localized cases may be distinguished by their bandstructures and Fermi surfaces which are depicted in Figures 2 and 3, respectively.

For the localized 4f-electrons there are 6 bands crossing the Fermi level. The 4f-electrons have a binding energy of 1.5 eV and do not contribute to the Fermi surface. In contrast, for itinerant 4f-electrons the band dispersion differs strongly around the Fermi energy and only 4 bands contribute to the Fermi surface. For ease of comparison between these two situations we use the same color code when labeling the bands. Due to the formation of flat 4f-electron bands a large contribution of the 4f-electrons to the density of states is observed around $E_F$. We note that a direct comparison of the density of states obtained in DFT to the large Sommerfeld coefficients observed in experiments [1, 2] would require a more elaborate account of the interaction of the f-electrons with the conduction electrons beyond DFT and the scope of this study. In contrast, the topology of the Fermi surface we focus on here is usually well-accounted for in DFT [24, 25].

Measurements that might be able to distinguish the itinerant and localized 4f-electron cases include ARPES and quantum oscillations, which each have their own merits for this compound. It should be noted at this point that quantum oscillation measurements require high crystal quality which would need to be improved upon as compared to the literature [16] giving values for the residual resistivity of 15 $\mu\Omega$cm for CeAuAl$_3$. As ARPES is limited to zero magnetic field it is mainly useful to observe the electronic structure of the itinerant 4f-electron. A transition from the paramagnetic to the helical state may be observable when cooling through the magnetic transition temperature. This may give further insight if the onset of magnetic order coincides with a change in the character of the 4f-electron from itinerant to localized. The Fermi surface of a localized 4f-electron in the field polarized state would be observable at large magnetic fields in quantum oscillation measurements. If further transitions, e.g., from the itinerant to localized state at intermediate magnetic fields or from a helical to the field polarized state will be observable in quantum oscillation measurements is hard to predict. This depends on the magnetic coupling for different magnetic field directions, the

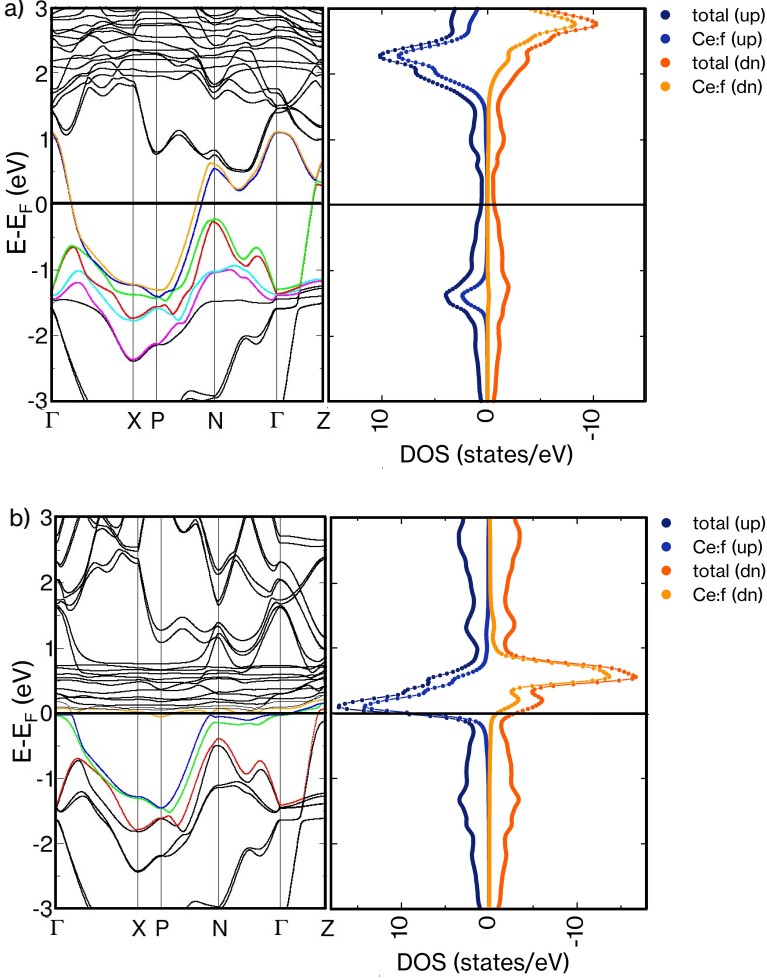

Figure 2: Band structure and density of states of CeAuAl$_3$: (a) Results assuming localized 4f-electron (LDA+U+SOC with $U$=4 eV), (b) Results assuming itinerant 4f-electron (LDA+SOC). The assignment of colors to the bands which cross the Fermi level corresponds to the assignment used for the Fermi surfaces shown in Figure 3.

effective masses of the quasi particles and the coupling between f-electrons and conduction electrons when changing the strength of the magnetic field.

In the following we show that the pronounced differences in the Fermi surface topology for the localized and itinerant 4f-electron visible in Figure 3a) and 3b) is also apparent when calculating the quantum oscillation orbits for both cases. This is illustrated in Figure 3c) and 3d) where we show the angular dispersion of the quantum oscillation branches in the lower panels. Three selected orbits are highlighted for localized f-electrons (green, red and cyan symbols) whereas 2 selected orbits (green and red) are highlighted for itinerant f-electrons. This selection of orbits may be traced for all magnetic field directions because they arise from the closed, singly-connected Fermi surface pockets shown in the upper panels of Figure 3c) and 3d), respectively. The Fermi surface sheet marked in cyan [band 43 (up)] of the localized 4f-electron vanishes in the case of itinerant 4f-electrons. Thus, no flat dispersing orbit at low frequencies would be observed in the quantum oscillation spectrum of the itinerant 4f-electrons. For the sheet marked in green [band 43 (dn)] the dispersion remains similar but the size of the orbits is reduced. The sheet marked in red [band 44 (up)] distinctly changes its

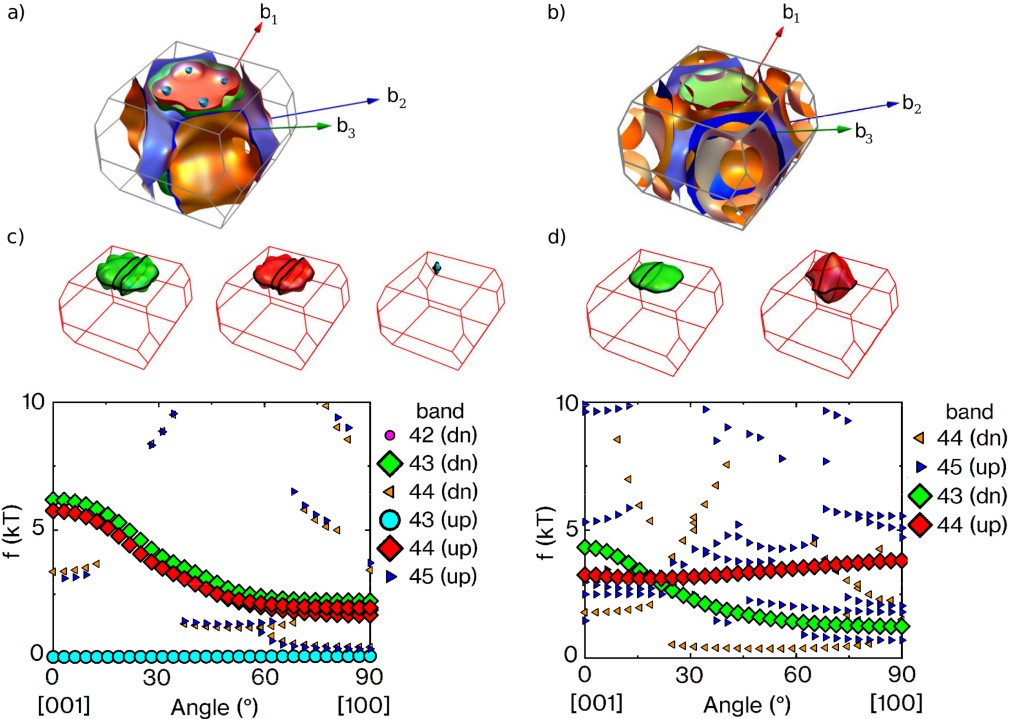

Figure 3: Fermi surfaces and quantum oscillation spectra of CeAuAl$_3$: a) Fermi surface for a localized 4f-electron comprising 6 sheets (LDA+U+SOC with $U$=4 eV). b) Fermi surface for an itinerant 4f-electron comprising 4 sheets (LDA). c) Characteristic orbits and quantum oscillation spectrum for a localized 4f-electron (LDA+U with $U$=4 eV). d) Characteristic orbits and quantum oscillation spectrum for an itinerant 4f-electron (LDA). The labelling of the Fermi surface sheets and orbits is shown in the legends of panels c) and d). The same colors are used for the same bands, e.g., band 44 (up) for the localized 4f-electron is colored in red as is band 44 (up) for the itinerant 4f-electron.

shape and hence the dispersion of the orbit. Furthermore, the orange [band 44 (dn)] and blue [band 45 (up)] sheets grow in size and are more complex when going from the localized to the itinerant configuration. In turn, we expect additional frequencies in the quantum oscillation spectra of itinerant 4f-electrons.

We conclude that distinct features of the Fermi surface and of the quantum oscillation spectra are unique to the localized or itinerant f-electrons, allowing to distinguish the cases experimentally.

## 4 Conclusions

The structural and magnetic properties of CeAuAl$_3$ were calculated using density functional theory. Assuming a localized 4f-electron picture by the DFT+U scheme, our results are in good agreement with available experimental data. Predictions of the electronic structure and Fermi surfaces for the localized and the itinerant 4f-electron case show that these may be readily distinguished in ARPES and quantum oscillation measurements.

# Acknowledgements

We wish to thank A. Bauer, P. Cermak, D. Eckert, C. Franz, A. Schneidewind, and M. Stekiel for discussions.

**Funding information** We gratefully acknowledge financial support by the DFG in the framework of TRR80 (project id 107745057), SPP 2137 (Skyrmionics) under grant no. PF393/19 (project id 403191981), DFG-GACR project WI 3320/3-1, ERC Advanced grant 788031 (ExQui-Sid), and Germany's excellence strategy EXC-2111 390814868.

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
