# Peer review of "Electronic structure of CeAuAl3 using density functional theory"

_SciPost Physics Proceedings, doi:SciPost Phys. Proc. 11, 008 (2023)_

## Round 1 · Referee Report · Anonymous · 2022-10-30

Strengths

1- thorough first ab initio calculations of the electronic structure in CeAuAl3

2- by comparing different calculation strategies and comparing the results to experiments, the authors are able to draw conclusions regarding the appropriate description of this material

Weaknesses

No significant weaknesses

Report

This is an interesting manuscript which deserves to be published in SciPost without major corrections. I learned something, both about the material and about the computational techniques.

Some minor suggestions follow below.

Requested changes

1- In the abstract, the Sommerfeld coefficient is given as 200mJ/molK^2, which strikes me as low given the very low T_K also given in the abstract. This is a contradiction, on the surface. However, when you look at the raw data in the paper by S. Paschen, it becomes clear that extracting the Sommerfeld coefficient is somewhat ambiguous because of the low-lying phase transition. I would have taken C/T just above T_N, which is much larger. The paper by Adroja also suggests this, and in fact your manuscript alludes to this discrepancy later on. I suggest leaving out the C/T value in the abstract, if a more realistic value cannot be justified, and then maybe discussing this just slightly more carefully lower down in the paper, when the current comparison with Adroja's value is made.

2- The possibility of distinguishing between the Fermi surface scenarios by quantum oscillation measurements is raised several times in the paper. However, this looks all but impossible: firstly, the residual resistivity is of order 50 microOhmcm, nearly two orders of magnitude to high to measure QO in laboratory magnets. Unless the authors have much, much better samples, this isn't looking promising. But more importantly, the system responds very strongly to magnetic field, so one would have to measure at fields of order Tesla in order to probe the low-T state that is being examined in the calculations, making the requirement on rho_0 even more stringent . The idea is good, but I think it would be useful for the reader if these challenges were also mentioned.

3- It wasn't so clear to me how the calculation underlying Figure 1 was made - is this something you can do within ELK without making large supercells? It looks like it, and it may be good to just mention this in the figure caption.

4- The third sentence starting from the bottom in section 3 ('It is finally possible...') sounds a bit unclear. Please check whether you can rephrase this and make it clearer.

---

## Round 2 · Author Response

we wish to thank the referee for her/his thoughtful comments and positive recommendation which helped us greatly to improve the manuscript. We followed all points raised by the referee and adapted the manuscript accordingly. We hope that the revised manuscript is now suitable for publication.
Sincerely,
André Deyerling (on behalf of the authors)
Below, we give a point-by-point response to all recommendations made by the referee.
"1- In the abstract, the Sommerfeld coefficient is given as 200mJ/molK^2, which strikes me as low given the very low T_K also given in the abstract. This is a contradiction, on the surface. However, when you look at the raw data in the paper by S. Paschen, it becomes clear that extracting the Sommerfeld coefficient is somewhat ambiguous because of the low-lying phase transition. I would have taken C/T just above T_N, which is much larger. The paper by Adroja also suggests this, and in fact your manuscript alludes to this discrepancy later on. I suggest leaving out the C/T value in the abstract, if a more realistic value cannot be justified, and then maybe discussing this just slightly more carefully lower down in the paper, when the current comparison with Adroja's value is made."
We reply: We thank the referee for raising this point. We agree that a more detailed discussion of the ambiguities involved is beneficial for the reader. We have removed the single value for the Sommerfeld coefficient from the abstract and discuss the analysis and the range of possible values for the Sommerfeld coefficient more carefully in the main text.
"2- The possibility of distinguishing between the Fermi surface scenarios by quantum oscillation measurements is raised several times in the paper. However, this looks all but impossible: firstly, the residual resistivity is of order 50 microOhmcm, nearly two orders of magnitude to high to measure QO in laboratory magnets. Unless the authors have much, much better samples, this isn't looking promising. But more importantly, the system responds very strongly to magnetic field, so one would have to measure at fields of order Tesla in order to probe the low-T state that is being examined in the calculations, making the requirement on rho_0 even more stringent. The idea is good, but I think it would be useful for the reader if these challenges were also mentioned."
We reply: We thank the referee for this fair comment. We adapted the manuscript by first pointing out the experimental challenges involved in the work program on quantum oscillations we propose here and second by putting more emphasis on other experimental techniques that are useful in band structure and Fermi surface determination such as angular resolved photoemission spectroscopy (ARPES) and angular correlation of positron annihilation radiation (ACAR). Finally, we briefly discuss possibly approaches to improve the sample quality.
"3- It wasn't so clear to me how the calculation underlying Figure 1 was made - is this something you can do within ELK without making large supercells? It looks like it, and it may be good to just mention this in the figure caption."
We reply: We thank the referee for this suggestion, which will improve the readability and clarity of the manuscript. In the calculations shown in Figure 1 large supercells were avoided by using a generalized version of the Bloch theorem, which combines translation symmetry and rotation symmetry in spin space [1]. This approach allows the calculation of (incommensurate) spiral spin structures within the chemical unit cell. In ELK and other electronic structure codes the implementation of this approach is called a ”spin spiral”-calculation [2]. We added an explanation and corresponding citations the the caption of Figure 1. We also updated figure 1. Now the data points correspond to a denser k-mesh of 27x27x20 instead of 18x18x7 which means better convergence.
"4- The third sentence starting from the bottom in section 3 ('It is finally possible...') sounds a bit unclear. Please check whether you can rephrase this and make it clearer."
We reply: We rephrased this part to make the discussion of large (itinerant 4f-electron) and small Fermi surface (localized 4f-electron) scenarios within the magnetic phase diagram clearer.
[1] L. M. Sandratskii, Symmetry analysis of electronic states for crystals with spiral mag-
netic order. I. general properties, Journal of Physics: Condensed Matter 3, 8565 (1991),
doi:10.1088/0953-8984/3/44/004.
[2] The Elk code, https://elk.sourceforge.io/elk.pdf.

---

## Round 2 · List of Changes

- We removed the single value of the Sommerfeld coefficient in line 2 of the Abstract.
- We added two sentences about the ambiguities in measuring the Sommerfeld coefficient of CeAuAl3 in section 3.1.
- We added further detail information to the caption of Figure 1.
- We updated Figure 1 with data points corresponding to calculations on a denser k-mesh
- We added a short paragraph in section 3.2, discussing the feasibility of quantum oscillations and ARPES measurements on CeAuAl3
- We moved the content of Appendix A to section 2
- We corrected minor errors and improved the English where necessary throughout the text
- We improved the layout

---

## Editorial Decision

published